

# Beneath the arctic greening: Will soils lose or gain carbon or perhaps a little of both?

Jennifer W. Harden[1,2], Jonathan A. O'Donnell[3], Katherine A. Heckman[4], Benjamin N. Sulman[5], Charles D. Koven[6], Chien-Lu Ping[7], Gary J. Michaelson[7]

[1]Stanford University, Palo Alto, California, 94301, USA

[2]U.S. Geological Survey, Menlo Park, California, 94025, USA

[3]Arctic Network, National Park Service, Anchorage, Alaska, 99501, USA

[4] Northern Research Station, U.S. Forest Service, Houghton, Michigan, 49931, USA;

[5]Environmental Sciences Division, Oak Ridge National Laboratory, Oak Ridge, Tennessee, 37830, USA

[6]Lawrence Berkeley National Lab, Berkeley California, 94720, USA

[7]School of Natural Resources and Extension, University of Alaska Fairbanks, Palmer, Alaska, 99645, USA

*Correspondence to:* Jonathan A. O'Donnell (jaodonnell@nps.gov)

Notice: This manuscript has been authored by UT-Battelle, LLC, under contract DE-AC05-00OR22725 with the US Department of Energy (DOE). The US government retains and the publisher, by accepting the article for publication,

acknowledges that the US government retains a nonexclusive, paid-up, irrevocable, worldwide license to publish or reproduce the published form of this manuscript, or allow others to do so, for US government purposes. DOE will provide public access to these results of federally sponsored research in accordance with the DOE Public Access Plan (http://energy.gov/downloads/doe-public-access-plan).


**Abstract.** Ecosystem shifts related to climate change are anticipated for the next decades to centuries based on a number of conceptual and experimentally derived models of plant structure and function. Belowground, the potential responses of soil systems are less well known. We used geochemical steady state models, soil density fractionation, and soil radiocarbon data to constrain changes in soil carbon based on measurements from detrital ("free light"), aggregate-bound ("occluded") and

complexed or chemically bound ("mineral associated") carbon pools and for bulk soil. We explored a space-for-time sequence of soils along a cold-to-warm climatic gradient from Alaskan Black Spruce forest soil with permafrost (Gelisols; 50 cm Mean Annual Temperature -1.5 ℃), Alaskan White Spruce forest soil lacking permafrost (Inceptisols; 50 cm MAT +3 ℃ ),  and



Iowa Grassland soil lacking permafrost (Mollisols; 50 cm MAT +9 ℃) developed on similar geologic substrates (wind-blown loess deposits). These temperature ranges were also representative of temperatures at 50 cm soil depth from model output by the Community Land Model for the years 2014, 2100, and 2300 for Interior Alaska. Fitting an exponential equation to depth trends in soil C down to 2 m depths, we found that depth distributions of organic C were related mainly to depths of rooting

and changes in bulk density. Using output from the geochemical steady state model, the direction and magnitude of the C loss or gain upon ecosystem shift was dictated by the C stocks of initial and final ecosystems. Radiocarbon measurements specific to each soil fraction (free light, occluded, and mineral associated) allowed us to constrain the timing of the potential loss or gain of C in each fraction driven by climatic shifts. Thawing from the Gelisol to Inceptisol in loess parent materials from present day to year 2100 resulted in small net gains to soil C, reflecting the net balance between loss of detrital and gain into

occluded and mineral associated C. Greater warming and shifts from Inceptisol to Mollisol analogous to predicted warming from circa 2100 to 2300 resulted in net C losses from both occluded and mineral associated C, although small gains to the free light C fraction occurred throughout the depth profile. Gains to occluded and mineral associated C post- thaw likely reflect aggregate formation and physical protection of C as well as formation of organo-mineral compounds that accompany microbial processing. Greater warming and shifts from Inceptisol to Mollisol, which are analogous to predicted warming circa 2100 to

2300, resulted in net C losses from both occluded and mineral associated C resulting from enhanced decomposition, small gains to the free light C fraction occurred throughout the transition to Mollisol reflecting deeper rooting of the tallgrass prairie system.

## 1 Introduction

Climate, land use, and land-cover change drive ecosystem shifts, with accompanying changes to aboveground and

belowground carbon (C) pools (Dixon et al., 1994; Pan et al., 2011). Detection of changing aboveground dynamics associated with ecosystem shifts has improved dramatically through recent conceptual and quantitative advances in modeling (e.g., Bonan et al., 2002; Grace et al., 2016), field measurements, (e.g., production, longevity, dispersal, allocation; e.g., Cleveland et al. 2015), relational trends with various metrics (e.g., allometric equations; Pan et al., 2011; Chave et al., 2014), and an increasingly robust set of spatiotemporal data (LANDSAT, Spectral satellite imagery; e.g., McDowell et al., 2015 and citations

therein). Changes in the soil that resides beneath the plant community, however, remain much less well understood with respect to conceptual (e.g., taxonomy; soil state and controlling factors; Bradford et al. 2016) and mathematical approaches (e.g., first order decomposition as in Todd-Brown et al., 2013). Further, there are fewer measurements of soil characteristics and relational trends to those measurements, and an inability to remotely sense soil carbon dynamics separately from aboveground dynamics. Although process-based models depicting soil physics and carbon dynamics are increasingly detailed, mechanistic, and

complex (Sulman et al., 2018; Wieder et al, 2015), confidence is relatively low with respect to modeling or forecasting changes in soil (Todd-Brown et al. 2013; Luo et al. 2016). The depth dependency of soil C storage and turnover remains particularly problematic because fewer measurements are available for deeper soils (Hugelius et al. 2014), which further hampers our



ability to link aboveground to belowground C, and also because the model uncertainty of soil physical climate increases with depth in the soil (Luo et al. 2016).

In many studies, soil C budgets are modelled as a steady-state system in which inputs are balanced by losses over the mean residence time (MRT) of the carbon for a given depth (Luo et al. 2016). Stocks and MRTs of soil C have been shown to vary

laterally and with depth according to a variety of environmental and substrate-specific factors that often cluster or covary within a given ecosystem (Davidson and Janssens 2006). For example, in boreal forests, C inputs (e.g., net primary production, dissolved organic C) to soils vary with depth according to depth distributions of moss, litter, and roots and their nutrient stoichiometries (Harden et al., 2012; Clemmensen et al., 2013; O'Donnell et al., 2016). Much like inputs, soil C losses (via decomposition, erosion, and lateral transport) also vary according to ecosystem states (Guillaume et al. 2015; Jones et al.

2017). Moreover, high-latitude ecosystems and their soils undergo a variety of disturbances, including wildfire, thermokarst, and thermal erosion, that vary in their return intervals and spatial extent, all of which impact the long-term C budgets of soils (e.g., O'Donnell et al. 2011a), and few of which are represented in most Earth system models.

MRTs and soil C storage metrics (gC m$^{-2}$ for given depth intervals) provide a constraint for the timing, direction and magnitude of change that occurs during ecosystem shifts. Radiocarbon ($^{14}$C) has proven to be a powerful tool for quantifying the MRT of

soil C (Trumbore 2000; He et al. 2016). Thus, given a simple postulation for an ecosystem shift from one state to another, depth distributions of MRT and C stock data offer a simple tool for constraining the direction and timing of changes in belowground C that accompany that ecosystem shift.

Soil C stocks in high-latitude regions are sensitive to climate- and disturbance-driven ecosystem shifts, which are often initiated by wildfire and/or permafrost thaw (Turetsky et al. 2012; Schuur et al. 2015). These vulnerabilities represent an uncertain and

potentially large feedback to climatic warming (Schuur et al., 2015). Given the rapid pace of warming in the Arctic, we targeted permafrost soils for study and asked which types of ecosystems and soils might represent new states for future, warmer climates. We used a space-for-time substitution approach to track long-term (decadal, century, and millennial scale) changes in soil C dynamics during transitions from frozen conditions to a warmer state. To minimize the influence of environmental and geologic factors, we limited our comparison to soils developed in late Pleistocene wind-blown sediment (loess) that include

a Gelisol from Interior Alaska, an Inceptisol from South-central Alaska, and a Mollisol from Iowa. We hypothesized that under warming conditions, the present-day Gelisols (soils with perennially frozen ground, or permafrost) could eventually manifest as Inceptisols (no recent permafrost) or ultimately as Mollisols (permafrost-free). Thus, treating these soils as stations along a long-term potential trajectory of warming allowed us to conduct a space-for-time substitution and investigate potential changes in soil C stocks driven by changes in both climate and vegetation over long time spans. Using this space-for-time approach,

we asked two questions. First, following a stepwise shift from a black spruce (*Picea mariana*)- and permafrost-dominated ecosystem (Gelisol) to a recently thawed white spruce (*P. glauca*) ecosystem (Inceptisol), *how much belowground C would be lost or gained by 2100?* Second, *how much additional C would be lost or gained following a further shift from white spruce to*




*grassland (Mollisol) by 2300*? In addition, we ask *how would distributions of belowground C shift among particulate, occluded, and mineral-associated fractions change under these transitions?* To address these questions, we compiled soil C stock and $^{14}$C profiles from the three soil types, and used a simple exponential equation to model depth distributions of soil C cycle parameters (C content (%), C density (g cm$^{-3}$), MRT). Using these depth-dependent distributions, we then simulated

changing soil C storage and fluxes associated with changing ecosystems following methods of Rosenbloom *et al*. (2006).

## 2 Methods

### 2.1 Study Sites

Soil samples were collected and analyzed from three study sites underlain by Pleistocene-aged loess silt, including a Gelisol in interior Alaska, an Inceptisol in south-central Alaska, and a Mollisol in Iowa. Parent material is one of the primary soil-

forming factors known to influence soil properties and state (Jenny 1941). Other soil forming factors (climate, vegetation) varied across sites, providing a means to test the effects of changing climate and ecosystem state on soil C storage and flux. For example, Iowa Mollisols formed with little or no permafrost throughout most of the depositional history of loess accumulation and organic matter burial. Alaska Inceptisols formed with no permafrost since at least 5,000 y BP as evidenced by a 5 ka volcanic tephra at ~1 m depth that was not deformed by frost heave; and Alaska Gelisols formed with continuous

permafrost since the Pleistocene.

The Gelisol profiles were collected in 2007 from mature black spruce stands near Hess Creek, approximately 150 km north of Fairbanks, Alaska (65.56758ºN, 148.92488ºW ; O'Donnell et al. 2013). All sampling locations were located on north-facing slopes, were somewhat poorly drained, and were underlain by ice-rich permafrost. The region is characterized by a continental climate, with temperature extremes ranging from -50 °C in winter to 35 °C in summer. Mean annual precipitation averages

270 mm, most of which falls during the summer growing season. In mature black spruce stands, the forest understory is composed of small woody shrubs (e.g., *Vaccinium vitis-idaea*, *Ledum groenlandicum*), feather mosses (*Pleurozium schreberi*, *Hylocomium splendens*) and reindeer lichens (*Cladina stellaris*, *C. arbuscula*). Organic-soil horizons composed of live moss, fibrous, and amorphous horizons that overlie mineral soils, and often exceed 20 cm in thickness (Harden et al. 2006; O'Donnell et al. 2011a). Active layer thickness averaged 45 ± 8 cm at mature black spruce sites (O'Donnell et al. 2011b). Permafrost

development at Hess Creek occurred in conjunction with loess deposition (i.e., syngenetic permafrost aggradation; French and Shur 2010), and is commonly referred to as *yedoma* in the literature (e.g., Strauss et al. 2013).

The Inceptisol profiles were mapped and correlated as the Bodenburg series. Samples for radiocarbon were collected on 12 June, 2012 by Gary Michaelson and C.L. Ping north of Palmer, Alaska (Clark et al, 2002) following profile C1701F91-1, GPS position: Lat. 61.63194 ºN; Long. 149.17556 ºW; Elev. 144 m. Inceptisols developed primarily in loess deposits

originating from the Matinuska and Knik glaciers (Muhs et al. 2004). Today, mean annual air temperature at Palmer is 2.7 °C



and mean annual precipitation is 302 mm. Sampling sites were situated in the southernmost portion of Alaska's discontinuous permafrost zone (Jorgenson et al. 2008), but permafrost was generally not present in this region. Study sites were dominated by white spruce with some mixed forest, with feathermoss-dominated forest floors generally less than 10 cm thick.

Mollisol samples were collected in 1997 at the Dinesen Prairie site, a 20-acre hillslope located near the town of Harlan, Iowa
5   (41.709 N, -95.281 W; Manies et al. 2001; Harden et al. 2002). Soils formed in loess deposits originated from glacial to post-glacial outwash along the Missouri River and distal loess sources in Nebraska (Bettis 1990; Muhs and Bettis 2000). Vegetation is dominated by tallgrass prairie plants, and the soils typically have very deep, dark A mineral-soil horizons with no organic horizons. Mean annual temperature in the region is 8.9°C and mean annual precipitation averages 850 mm.

## 2.2 Climate Analyses

10   We used CLM4.5 predictions of soil temperatures at 50 cm to estimate potential site analogues for expected changes in climate over the next few centuries (Fig. 1a), as driven by the high-emissions RCP8.5 scenario and described in Koven et al. (2015). Measured mean monthly temperatures at 50 cm depths (Fig. 1b) illustrate dramatic differences in thaw season lengths, which are comparable to future warming scenarios for Interior Alaska over the next century (Fig. 1a).



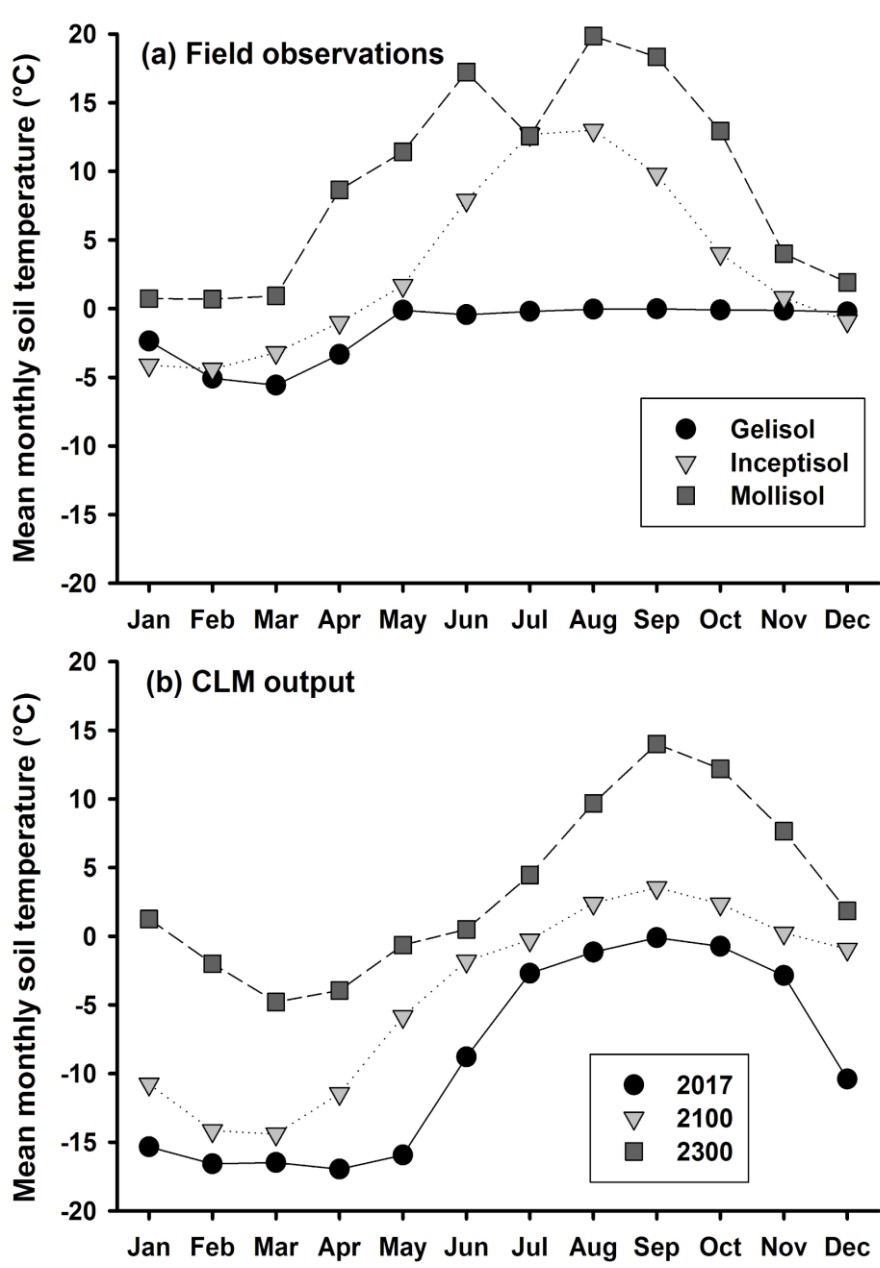

**Figure 1.** a) Modelled monthly soil temperatures at 50-cm depths below organic/mineral soil boundary for circa 2014, 2100, and 2300 using the Community Land Model. b) Measured 50cm soil temperatures for 3 ecosystems: Alaska Blackspruce (Gelisol); Alaska Whitespruce (Inceptisol); Iowa Tallgrass Prairie (Mollisol).



## 2.3 Soil analyses

Soils were described according to USDA-Natural Resources Conservation Service field manual (Schoeneberger et al., 2012) and to U.S. Geological Survey (USGS) protocols for boreal soils (Manies et al., 2004). Root abundance was calculated for each soil horizon following methods of the soil development index (Harden 1982), in which categories for very few, few,

common, and many roots for each category of very fine, fine, and medium roots were assigned values of 5, 10, 20, and 30 points, respectively. Field parameters for bulk soil horizons and fractionated soil samples are reported in Table S1 and S2, respectively.

We used different techniques for determining C and N content of soil samples. Alaskan Gelisol and Iowa Mollisol samples were analyzed for total C at the USGS laboratories in Menlo Park, CA, by high temperature combustion using a Carlo Erba

NA1500 elemental analyzer (CE Elantech Inc., Lakewood, NJ) or by measuring the carbon dioxide ($CO_2$) produced by combusting the sample in a stream of oxygen ($O_2$) using a Fisons NA1500 elemental analyzer (EA)/ Optima isotope ratio mass spectrometer (IRMS). For Gelisol samples, carbonates were removed prior to combustion using acid fumigation techniques (see O'Donnell et al. 2011 for details). Mollisol samples were also analyzed for inorganic C by measuring the $CO_2$ generated by heating a sample at 105 °C in acid using a UIC coulometer. Organic carbon (C) was calculated as the difference between

total and inorganic C for Mollisol samples. For Alaskan Inceptisol samples, organic C was determined at the University of Alaska Fairbanks (UAF) Palmer labs using LICO CNH Carbon analyzer to measure the total C after pretreatment with HCL to remove inorganic C.

One profile from each study region was selected for density fractionation (Strickland and Sollins 1987, Swanston et al. 2005). Briefly, soils were air dried, sieved to <2 mm and density-separated to 1.65 g cm$^{-3}$. Floating organics following gentle shaking

represent the "free/light fraction"; floating organics following mechanical mixing (1 minute with Polymix benchtop mixer set at 1500 rpm) and sonication (1500 J per gram of soil) represent "occluded fraction"; heavy organics at the end of shaking and sonication represent "mineral-associated fraction".

Bulk and fractionated organic C were analyzed for radiocarbon ($^{14}$C) content to estimate the MRT and average $^{14}$C age of

organic C fractions. Samples were graphitized at the USGS $^{14}$C prep lab in Reston, Virginia. Samples were combusted at 900°C for 6 hours with CuO and Ag in sealed quartz test tubes to form $CO_2$ gas. The $CO_2$ was then reduced to graphite through heating at 570°C in the presence of $H_2$ gas and a Fe catalyst (Vogel et al., 1987). The $^{14}$C abundance of each graphitized sample was measured at the Center for Accelerator Mass Spectrometry, Lawrence Livermore National Lab (Davis et al., 1990) or at the KECK lab at UC Irvine. Reported fraction modern (F$^{14}$C) and $\Delta^{14}$C values include a background subtraction determined

from $^{14}$C-free coal or wood and a $\delta^{13}$C correction to account for isotopic fractionation (Stuiver and Polach, 1977).



## 2.4 Statistical analyses and soil carbon modelling

Models used to calculate MRT from [14]C measurements assume that new organic C inputs to each fraction bear an atmospheric [14]C signature of a given year and are mixed into the pool or fraction according to a time-dependent steady-state model (Trumbore, 1993; Torn et al., 2002). MRT of light ($MRT_L$), occluded ($MRT_O$), and mineral associated fractions ($MRT_M$)

respectively were calculated independently.

Depth-attenuation of organic C content (%C), C density (gC cm$^{-3}$), and MRT for bulk soil (years) was modeled after Rosenbloom et al. (2006) in which surface C ($C_s$) declines exponentially with depth (Eq. 1). For each soil profile within each soil type, we modeled the depth distribution of the three C parameters (C content, C density, C MRT) following the equation from *Rosenbloom et al.* [2006]:

$$C(z) = C_s \left[ e^{\left(\frac{-z}{Z^*}\right)} \right] \qquad (1)$$

where *C(z)* is C parameter value at depth *z*, $C_s$ is the C parameter value at the surface, and $Z^*$ is an empirical depth scaling parameter. $C_s$ and $Z^*$ were optimized for each C parameter and soil profile using the Solver function in Microsoft Excel. Specifically, we used Solver to optimize the relationship between the log of observed soil profile C parameters and the log of C(z) (via Eq. 1). Fits for Eq. 1 (Fig. S1) used measurements from mineral soil layers (%C ≤ 20%) and fractionated soil

from the uppermost mineral horizon ($C_s$) to the depth ($Z_{min}$) where the C parameter reached its minimum value for the profile ($C_{min}$). In all Gelisols as well as several non-Gelisol profiles, $C_{min}$ occurred above the bottom of the profile and C content increased in deeper layers. Therefore, we averaged C parameter values at depths below $Z_{min}$ to estimate mean parameter value for deep soil layers ($C_{deep}$). Output for $R^2$, *P*-value, slope, $C_{min}$, $C_{deep}$, $C_s$, and $Z^*$ were recorded for each fit. We compared Solver results against optimizations using the curve-fit nonlinear regression function in scipy (Jones et al, 2001; see also Fig.

S2). Regression between Solver and Python scripts indicated agreement for all four parameters.

Temporal transitions of soil C during ecosystem shifts were modeled using depth-dependent C stocks and decomposition coefficients in each of the Free-light, Occluded, and Mineral-Associated fractions (calculated using Eq. 1) to address our primary research questions. In the model, ecosystem transitions from black spruce (thaw and warming) to white spruce or from white spruce (warming) to grassland occurred instantaneously (i.e., stepwise) toward the new ecosystem state. Soil C stocks

were assumed to start at the previous ecosystem state and then evolved toward the new ecosystem state at rates determined by the stocks and MRT of the newer, warmer ecosystem and soil. Soil C parameters were calculated in 20-cm increments and changes were calculated at 10-year time steps following Rosenbloom et al., (2006; Eq. 2):

$$C^{t+1}_{actual(z)} = C_{equil(z)} - \left[ C_{equil(z)} - C^{t}_{actual(z)} \right] * \left[ e^{-dt/TT_z} \right] \qquad (2)$$



where $C_{equil(z)}$ (in gC m$^{-2}$) is the storage potential of layer $z$ (i.e., observed C stocks for the new ecosystem state, for example observed C stocks for the Inceptisol were used to calculate new storage potentials for a thawing Gelisol); $C^{t}_{actual(z)}$ (gC m$^{-2}$) is the C inventory at time $t$, and $C^{t+1}_{actual(z)}$ is the C storage at the next time step (i.e., $t + 1$), and $TT_Z$ is the MRT at depth $z$. This calculation was applied separately for each soil C fraction (free light, occluded, and mineral-associated) and depth using

the appropriate depth-resolved C stocks and MRTs calculated from Eq. 1.

## 3 Results

### 3.1 Soil C and radiocarbon profiles of soil fractions

C percent declines with depth in bulk soil, as captured by the depth parameters for Eq. 1 (Table 1). Parameters of $Z^*$ are deeper for Gelisols and Inceptisols than for Molliols, whereas $Z_{min}$ parameters are shallower for Gelisols and Inceptisols than

Mollisols. $C_{deep}$ values are greatest for Gelisols, followed by Inceptisols and then Mollisols.

The $\Delta^{14}$C content of soil organic matter and its depth trend reflect the influence of both aging (e.g., the time since photosynthesis fixed the C (Gleixner, 2013) and turnover (e.g., C inputs, losses, and retention). In Iowa Mollisols, the F$^{14}$C of the light fraction is relatively modern at all depths and reflects rapid incorporation of the atmospheric $^{14}$C signal within a few

years of sampling (Fig. 2, S). For example, the atmosphere in 1997 had a F$^{14}$C of approximately 1.1 to 1.15 (Hua et al. 2013), and the light fraction values of the Mollisol ranged from F$^{14}$C 1.07 to 1.19 (Fig. 2, S1). F$^{14}$C of occluded and mineral-associated fractions within the Mollisol decreased with increasing depth. F$^{14}$C for the light fraction of the Alaskan Gelisol and Inceptisol, in contrast to the Mollisol, reflected post-bomb values only near the surface and decreased dramatically at depths below 60-80 cm (Fig. 2a-b, S1), reflecting the presence of substantially older C at depth. In the Gelisol and Mollisol, the mineral-

associated fraction was oldest (Fig. 2a, c) and in the Inceptisol, the occluded fraction was oldest (Fig. 2b).




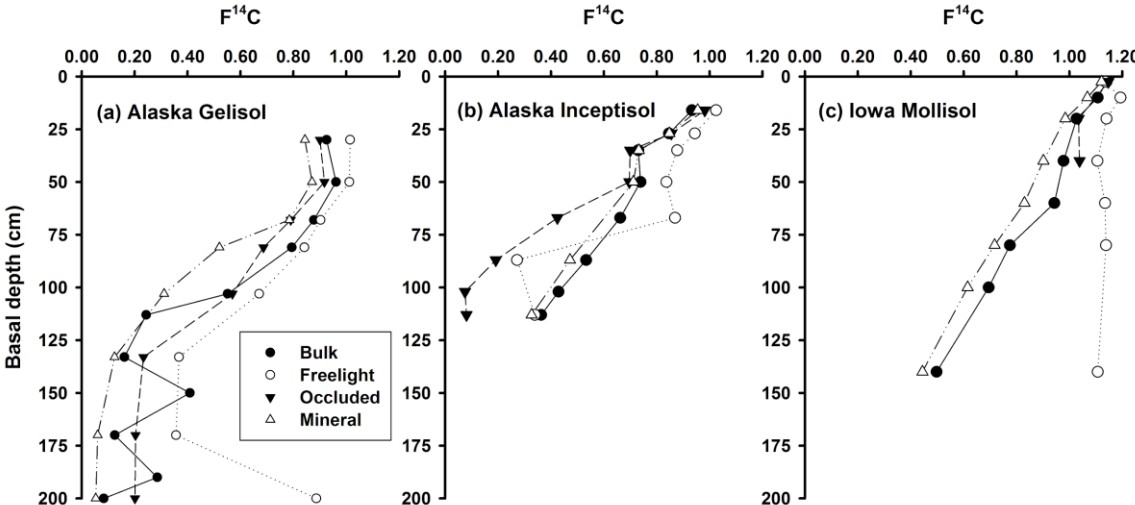

**Figure 2. Radiocarbon content of fractionated soil organic matter (reported as F$^{14}$C) in soils under black spruce (Gelisol), white spruce (Inceptisol), and grassland (Mollisol).**

### 3.2 Carbon content, density, and mean residence time of soil profiles

5   C density (g cm$^3$) declined with depth in all fractions (Fig. 3) and reached its minimum value ($C_{min}$) within two meters of the mineral-soil surface in all three soils. Values for $Z_{min}$, or the depth at which bulk soil C density first reaches its minimum value, were slightly deeper for Mollisols (137 ± 73 cm) than for the other soil types (56 ± 27 cm and 65 ± 35 cm for Gelisols and Inceptisols, respectively; Table S3).



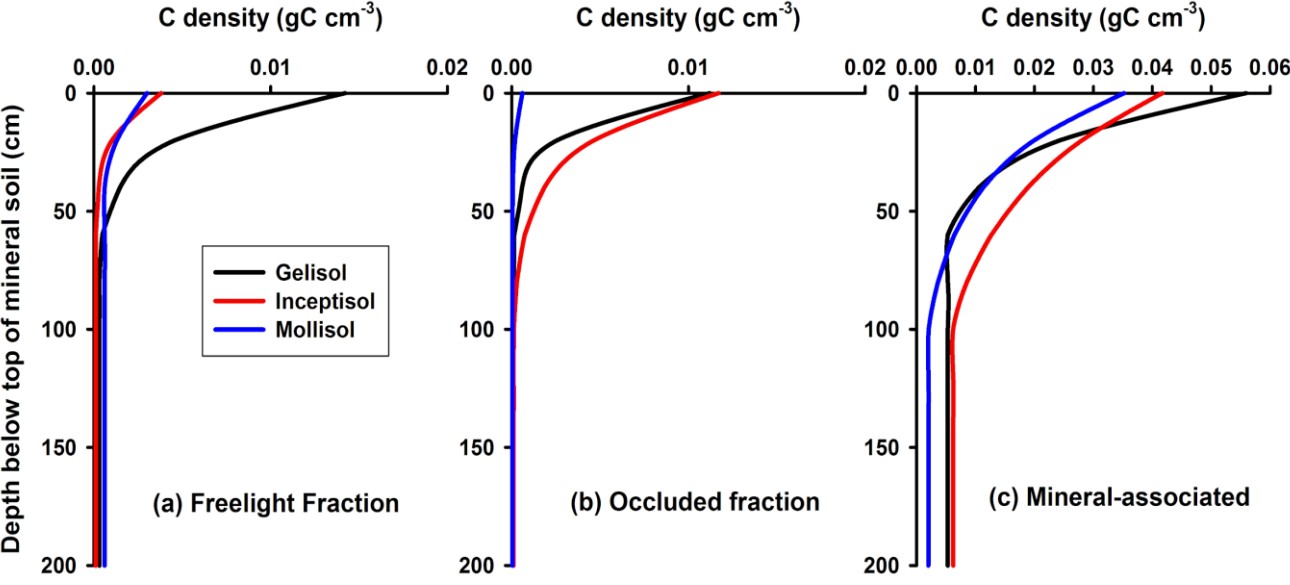

**Figure 3. Depth models for carbon stored in a fractionated soil from a Gelisol, Inceptisol, and Mollisol based on data-model fits from Eq. 1.**

5    Soil C content (%; Table 1), density (gC cm$^{-3}$; Fig. 3; Table S3) and decomposition coefficients ($k$, y$^{-1}$; Table S4) declined with depth. Although there was considerable variability and overlap among profiles and soil types, Gelisols generally had higher surface ($C_s$) densities. $Z^*$ and $Z_{min}$ values for Mollisols were greater than those for Gelisols and Inceptisols.

The decomposition rate constant $k$, calculated from the inverse of MRT of the bulk soil declined with depth in all three soil types (Table 2), with the most dramatic increases in $k$ values and depth occurring in the Gelisol. MRTs generally followed the

10    order Gelisol > Inceptisol > Mollisol. For bulk-soil data at 30 to 40 cm depth below the O horizons, the MRTs were approximately 500 y for the Mollisol, 3000 y for Inceptisol. and 2000 to 7000 y for the Gelisol. At 80 cm depth, the MRTs were about 4000, 7000, and 25,000 yrs respectively. At 130 to 140 cm depths, the MRTs differed by several millennia and ranged from 8200 (Mollisol), 19,000 (Inceptisol), to 55,000 y (Gelisol). Comparing the 30-40 cm to the 140 cm depths, the Gelisol $k$ slowed by a factor of 40 while the Inceptisol $k$ slowed by a factor of 6 and the Mollisol by a factor of 8.



### 3.3 Controls on soil C parameters

For bulk soil C content (%), we observed that $Z_{min}$ was highly correlated with the depth of maximum bulk density (Pearson coefficients: $R^2 = 0.599$; $P = 0.0003$; Table 1; see also Pearson statistics in Table S5) and with maximum rooting depth ($R^2 = 0.67$; $P = 0.0064$). $Z_{min}$ was also correlated with $Z^*$ ($R^2 = 0.86$; $P < 0.001$).

### 3.4 Modeling soil C during ecosystem change

The $Z^*$ values for C stored in free-light and occluded fractions were consistently shallower than for the mineral-associated fractions (Fig 3; Table S3), whereas the $Z_{min}$ depths were similar among fractions, at around 1 m depth or greater. The mineral-associated C stored below the depth of $C_{min}$ (as indicated by $C_{deep}$) was significantly greater than $C_{deep}$ of the other fractions in all soil types. $Z_{min}$ for the $k$ values of all fractions were consistently deeper for the Gelisols than the other soils. The $k$ value at the depth of $Z^*$ was consistently higher for Mollisols than the other soil types. The $k$ value at the mineral soil surface ($k_s$) for all fractions was consistently lower in Gelisols than in other soil types.

Model simulations of ecosystem change from the Gelisol to Inceptisol projected a net gain of 0.18 g C m$^{-2}$ to mineral soil by the year 2100 (Fig. 4a). The uppermost profile of the Gelisol was dominated by the faster-cycling free-light fraction and consequently lost C rather quickly (Fig. 4b), while the subsoil stabilized C into occluded and mineral-associated forms that were more prevalent in the Inceptisol relative to the Gelisol (Fig. 4a).




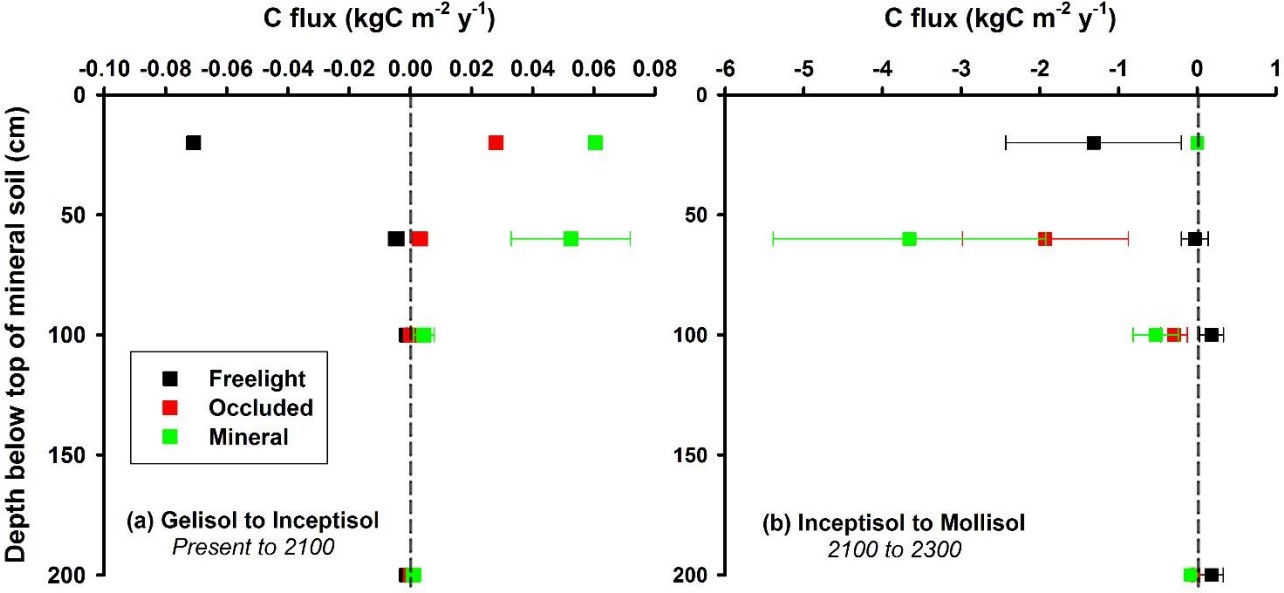

**Figure 4. Soil C flux estimated from a time model of ecosystem transitions from Gelisol to Inceptisol (circa 2000 to 2100) and Inceptisol to Mollisol (circa 2100 to 2300) using Eq. 1 output for carbon in fractionated soil from Suppl. Tables 1, 2. Fluxes are averaged for entire timecourse for each depth interval shown.**

Modeling the transition from Inceptisol to Mollisol during years 2100 to 2300 resulted in a net loss of 11 g C m$^{-2}$ of C from soil by the year 2300 (Fig. 4b). C loss from all three soil fractions reflected lower C storage in Mollisols than Inceptisols (Fig. 4b) for most soil depths. However, free-light fraction C increased at depths greater than 100 cm. Moreover, the loss rates were rapid, reflecting fast C turnover rates in Mollisols (Table 1, S4).

10 **4 Discussion**

Depth characterization of soil C by the formulation of Eq. 1 elucidates some important processes that may be generalizable and useful in regional analyses. Parameters of the depth relationship such as $C_{min}$, $Z_{min}$, $C_s$ and $Z^*$ likely tell us about multiple aspects of C cycling and may lend insight into the potential magnitude and timing of changes in belowground C during ecosystem transitions. A decline in decomposition coefficients (i.e., $k$ values) with depth reflects protection or stabilization of
15 C by both biotic and abiotic processes. Given the strong correlations we observed between $Z_{min}$ and maximum rooting depth, we postulate that $Z_{min}$ indicates a threshold above which carbon retention is dominated by biotic processes such as root input, microbial processing, and stabilization of C in the biotic medium (Creamer et al, in prep). Below Zmin, rooting no longer





dominates the depth attenuation of C storage and while biotic input and processing are present, carbon storage and retention are "quasi-biotic" with the influences of mineralogy, porosity, and hydrology equally prevalent.    In other studies, C storage and turnover were found to be closely coupled to surface area of clays and oxides, reaching a saturation level that changes with mineral transformations occurring on very long timescales (Lawrence et al 2015). Thus, $C_{min}$ and its depth $Z_{min}$ reflect the

potential for long-term preservation of C that is transformed only under new conditions that re-set $Z_{min}$ . For example, if Zmin is controlled mainly by rooting depth, then C below Zmin likely persists because of the paucity of root and microbial activity ; upon ecosystem shifts that enhance rooting depth, however, this carbon may be more readily accessible (Hicks Pries et al., 2018).

During the loss of permafrost and associated transition from the black spruce-dominated Gelisol to the white spruce-dominated

Inceptisol, we projected C losses from the uppermost mineral soil layers (Fig. 4) that were offset and outpaced by C gains in deeper horizons. C gains were generated from an increase in occluded (aggregate) fraction. This reflects the fact that $C_s$ for occluded C of the Gelisol (0.0112) was less than $C_s$ for occluded C of Inceptisol (0.0117), while $C_s$ for free light C was greater in the Gelisol (0.142) than the Inceptisol (0.0103). In deeper horizons represented at $Z_{min}$, we projected net C gains in the mineral associated fraction because $C_{min}$ of mineral-associated C in the Gelisol was less than that of Inceptisol (Table S3).

However, $C_{min}$ of free light fraction C was higher in the Gelisol than in the Inceptisol, leading to losses of free light C at depth during the transition. The surface and near-surface transitions in C stocks occurred relatively quickly and reached steady state by circa 2100 owing to faster turnover in detrital fractions at the surface. The gains into the aggregate- and mineral-stabilized pools of the occluded and mineral-associated fractions, respectively, occurred more slowly. The capacity for the occluded and mineral associated fractions to protect C is supported by the $^{14}$C-depleted values of these fractions in the Inceptisol (Fig. 2b).

Additionally, the occluded C fraction continued to act as a net C sink until at least circa 2300 (data not shown). It is important to note however that organic horizons likely would lose carbon as the ecosystem transitions from black spruce to warmer white spruce.. Based on data from this study, organic horizons of Gelisols averaged 0.87 +/- 0.50 and Inceptisols averaged 0.79 +/- .87 gC/cm$^2$. Thus thinning of O horizons could readily offset C gains into the mineral soil.

The transition from white spruce-dominated Inceptisol to grassland Mollisol suggests an overall net loss of C from soil,

although we projected gains to the free light fraction at depth (Fig. 4b) likely from deeper rooting of grassland plants (see also Harden et al, 2002). The loss of organic horizons, not included in these models, would also contribute another 0.04 g C m$^{-2}$ yr$^{-1}$ to the atmosphere.

Our results highlight the roles of different physical, chemical, and biological mechanisms in driving accumulation and persistence of soil C stocks, and illuminate the potential for these mechanisms to shift under changing climatic and ecosystem

states. For example, Gelisol soil C stocks were primarily preserved by low temperatures and frozen water, allowing the persistence of large free light fraction C stocks that became vulnerable to decomposition after warming. The alleviation of this thermal constraint following thaw and the subsequent transition to Inceptisol allowed for the loss of this relatively bio-labile



and bio-available fraction (Janzen et al., 1992). The shift to Inceptisol or Mollisol was accompanied by a shift in the dominant soil C stabilization mechanism toward physical occlusion in aggregates and bonding to mineral surfaces. These mechanisms have the potential to store large amounts of soil C, offsetting some of the losses of free light C due to thawing of permafrost, but are subject to different constraints on the capacity and turnover rates of soil C.

The results presented here are qualitatively in accord with ecosystem model simulations suggesting that increased productivity associated with both elevated $CO_2$ and warming in the permafrost region outpaces carbon losses from increased respiration on the multi-decadal scale, but that on the longer timescales beyond this century, losses from decomposition outweigh the gains (Koven et al., 2015; McGuire et al., 2017).

Several important caveats should be noted in this approach. We used a step-wise event for the ecosystem transition, meaning
that gradual changes such as warming and disturbance events such as wildfire are not specifically captured in this approach. Moreover, ecosystem responses to higher atmospheric $CO_2$ are not represented by soil or modern ecosystem analogs. Nevertheless, the combination of a mathematical approach to characterizing soil depth profiles with measurements of physical soil C fractions allows for a more robust constraint to potential soil C losses and gains compared to a more typical accounting confined to a given depth interval. A significant strength of this approach is that it integrates both abiotic (e.g., warming) and
biotic (e.g., changing plant communities) drivers of shifts in soil C without relying on the process-specific assumptions underlying ecosystem models. With this measurement-based approach, modern associations of ecosystems with soil type can be used to constrain soil C states for future ecosystem-climate analogs.

## 5 Conclusions

This study explored the potential for warming of permafrost soils (Gelisols) transitioning to Inceptisols and Mollisols to
sequester or release C to the atmosphere. We found that the transition from Gelisol to Inceptisol slightly increased total C stocks likely related to enhanced aggregation and organo-mineral bonding, while the transition from Inceptisol to Mollisol decreased soil C stocks owing to higher temperatures and enhanced decomposition. These metrics and methods have potential for constraining spatiotemporal changes in soil carbon that accompany ecosystem shifts using large datasets of soil depth profiles that report bulk density, organic C, and have some data or proxy information on C fractions and their turnover.
Without fraction data and estimates for their turnover times, these methods provide constraints for changes in belowground net C stocks upon ecosystem shifts, but the timing of changes must be constrained by other methods or assumptions. Our space-for-time approach integrates changes in vegetation, climate, and mineral factors to provide constraints on potential changes in soil C stocks without depending on the process-based assumptions underlying ecosystem biogeochemical models.

## 6 Code availability

In the manuscript supplement, we have provide a detail set of instructions for running the SOLVER optimization procedure in Microsoft Excel. We also provide Python code for fitting Eq. 1 to soil C profiles and for comparing SOLVER and Python output.

## 7 Data availability

Supplemental tables are included in supplemental link and are available through the International Soil Radiocarbon Database (https://international-soil-radiocarbon-database.github.io/ISRaD/) and the International Soil Carbon Network (http://iscn.fluxdata.org/).

## 8 Author contributions

Harden conceived the experimental design; Harden, Heckman, Ping, and Michaelson contributed samples for analysis; O'Donnell structured the Microsoft Excel programs; Koven provided community land model results; Sulman provided checks on excel programs by constructing model codes in Python, with oversights from Koven. All authors contributed insights to data synthesis, conclusions, and implications for future work.

## 9 Competing interests

Authors declare no conflict of interest.

## 10 Acknowledgments

We are grateful for the many discussions with Gustaf Hugelius, N. Rosenbloom, Yuji He, Claire Treat, Carlos Sierra, Kristen Manies, and Xiaomei Xu. Yuji He also contributed to Python coding. This material is based in part upon work supported by the U.S. Department of Energy, Office of Science, Office of Biological and Environmental Research (Sulman); National Park Service's Arctic Inventory and Monitoring Network (O'Donnell); USDA Forest Service (Heckman); USGS Climate and Land Use Change (Harden); Stanford Visiting Scholar Program (Harden).



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

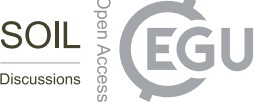



**Table 1. Model fits for the depth attenuation of %C in bulk soil and profile characterizations of horizons and rooting depths.**

| Soil type | Site ID | $Z_{adj}$ | $Z^*$ | $Z^*_{adj}$ | $Z_{min}$ | $Z_{min\_adj}$ | $C^*$ | $C_s$ | $C_{min}$ | Mean $C_{deep}$ |
|---|---|---|---|---|---|---|---|---|---|---|
| gelisol | HCCN 2 | 19.0 | 33.8 | 52.8 | 94.0 | 113.0 | 2.01 | 5.45 | 0.40 | 0.72 |
| gelisol | HCCN 3 | 25.0 | 29.2 | 54.2 | 56.0 | 81.0 | 1.87 | 5.07 | 0.93 | 1.52 |
| gelisol | HCCN 4 | 30.0 | 14.8 | 44.8 | 45.0 | 75.0 | 2.01 | 5.45 | 0.32 | 0.75 |
| gelisol | HCCN 5 | 25.0 | 17.5 | 42.5 | 32.0 | 57.0 | 1.00 | 2.73 | 0.94 | 1.34 |
| mollisol | DPPR 3/4 | 0.0 | 56.6 | 56.6 | 220.0 | 220.0 | 1.12 | 3.05 | 0.12 | 0.15 |
| mollisol | DPPR 2 | 0.0 | 35.5 | 35.5 | 40.0 | 40.0 | 1.79 | 4.86 | 1.58 | |
| mollisol | KH DPPR 2 | 0.0 | 37.1 | 37.1 | 100.0 | 100.0 | 1.79 | 4.86 | 0.25 | |
| inceptisol | C1701F91-2 | 0.0 | 35.7 | 35.7 | 113.0 | 113.0 | 3.20 | 8.69 | 0.32 | |
| inceptisol | Bodenberg | 0.0 | 9.8 | 9.8 | 29.0 | 29.0 | 6.05 | 16.42 | 1.25 | 1.28 |
| inceptisol | 59AK090002 | 8.0 | 15.9 | 23.9 | 36.0 | 44.0 | 0.99 | 2.68 | 0.33 | 0.31 |
| inceptisol | 59AK090001 | 8.0 | 17.4 | 25.4 | 53.0 | 61.0 | 0.98 | 2.66 | 0.18 | 0.33 |
| inceptisol | 59AK090003 | 8.0 | 14.5 | 22.5 | 38.0 | 46.0 | 0.97 | 2.63 | 0.20 | 0.25 |
| inceptisol | S03AK-090-010 | 16.0 | 21.3 | 37.3 | 81.0 | 97.0 | 1.73 | 4.70 | 0.23 | |
| inceptisol | S03AK-068-004 | 5.0 | 6.7 | 11.7 | 36.0 | 41.0 | 3.04 | 8.25 | 0.19 | 0.30 |
| inceptisol | S03AK-240-008 | 11.0 | 15.6 | 26.6 | 55.0 | 66.0 | 1.71 | 4.66 | 0.14 | 0.39 |
| inceptisol | 56AK170002 | 5.0 | 27.8 | 32.8 | 91.0 | 96.0 | 2.96 | 8.04 | 0.23 | |
| inceptisol | 91AK170002 | 0.0 | 47.1 | 47.1 | 111.0 | 111.0 | 2.25 | 6.12 | 0.71 | |
| inceptisol | 91AK170001 | 16.0 | 64.3 | 80.3 | 77.0 | 93.0 | 1.68 | 4.56 | 1.53 | 1.35 |
| inceptisol | 79AK170006 | 11.0 | 17.7 | 28.7 | 22.0 | 33.0 | 3.29 | 8.94 | 2.97 | 3.04 |
| inceptisol | 89AK170001 | 0.0 | 31.6 | 31.6 | 82.0 | 82.0 | 4.15 | 11.27 | 0.98 | 0.68 |
| inceptisol | SO4AK-176-001 | 18.0 | 60.3 | 78.3 | 140.0 | 158.0 | 0.88 | 2.39 | 0.32 | |





**Table 2. Mean resident time (MRT) of soil carbon estimated for bulk soil profiles and profiles of soil fractions.**

| Profile name | Layer name | Top depth (cm) | Bottom depth (cm) | Bulk | Free-light | Occluded | Mineral-Associated |
|---|---|---|---|---|---|---|---|
| HCCN2/3 | HCCN 2.30/HCCN 3.30 | 24 | 30 | 1660 | 291 | 1087 | 1660 |
| HCCN2/3 | HCCN 2.53/HCCN 3.50 | 30 | 50 | 1378 | 303 | 934 | 1378 |
| HCCN2/3 | HCCN 2.68 | 37 | 68 | 2364 | 1065 | 2293 | 2364 |
| HCCN2/3 | HCCN 3.81 | 53 | 81 | 7586 | 1680 | 3817 | 7586 |
| HCCN2/3 | HCCN 3.103 | 68 | 103 | 18060 | 4103 | 6239 | 18060 |
| HCCN2/3 | HCCN 2.113 | 93 | 113 | | | | |
| HCCN2/3 | HCCN 2.133 | 113 | 133 | 57568 | 14072 | 26795 | 57568 |
| HCCN2/3 | HCCN 2.150 | 133 | 150 | | | | |
| HCCN2/3 | HCCN 2.170 | 150 | 170 | 127833 | 14767 | 31929 | 127833 |
| HCCN2/3 | HCCN 2.190 | 170 | 190 | | | | |
| HCCN2/3 | HCCN 2.200 | 190 | 200 | 146218 | 1214 | 32345 | 146218 |
| Bodenburg_fractionated | C1701F91-2 . OA . 0-5cm . 07-02-14 | 0 | 5 | | | | |
| Bodenburg_fractionated | C1701F91-2 . A . 9-16cm . 07-02-14 | 9 | 16 | 636 | 250 | 458 | 636 |
| Bodenburg_fractionated | C1701F91-2 . Bw1 . 20-27cm . 07-02-14 | 20 | 27 | 1619 | 171 | 1537 | 1619 |
| Bodenburg_fractionated | C1701F91-2 . Bw2 . 28-35cm . 07-02-14 | 28 | 35 | 3097 | 1306 | 3601 | 3097 |
| Bodenburg_fractionated | C1701F91-2 . Bwb1/Ab . 43-50cm . 07-02-14 | 43 | 50 | 3408 | 1738 | 3655 | 3408 |
| Bodenburg_fractionated | C1701F91-2 . 60-67cm . 07-02-14 | 60 | 67 | | 1389 | 11119 | |
| Bodenburg_fractionated | C1701F91-2 . 80-87cm . 07-02-14 | 80 | 87 | 9177 | 21823 | 34359 | 9177 |
| Bodenburg_fractionated | C1701F91-2 . 95-102cm . 07-02-14 | 95 | 102 | | | 100284 | |
| Bodenburg_fractionated | C1701F91-2 . 106-113cm . 07-02-14 | 106 | 113 | 16817 | 15917 | 92693 | 16817 |
| DPPR2 | KH DPPR2.5 | 0 | 5 | 75 | 64 | 58 | 75 |
| DPPR2 | KH DPPR2.10 | 5 | 10 | 137 | 84 | | 137 |
| DPPR2 | KH DPPR2.20 | 10 | 20 | 434 | 62 | 212 | 434 |
| DPPR2 | KH DPPR2.40 | 20 | 40 | 1066 | 88 | 202 | 1066 |
| DPPR2 | KH DPPR2.60 | 40 | 60 | 1808 | 63 | | 1808 |
| DPPR2 | Heckman composite 80 | 60 | 80 | 3316 | 60 | | 3316 |
| DPPR2 | Heckman composite 100 | 80 | 100 | 5177 | | | 5177 |
| DPPR3 | Heckman composite 140 | 100 | 140 | 10232 | 87 | | 10232 |