# Peer review of "Beneath the arctic greening: Will soils lose or gain carbon or perhaps a little of both?"

_SOIL, 2018_

## Referee Comment (RC1) · Anonymous Referee #1 · 14 Feb 2019

The ongoing climate change is affecting permafrost soils and the carbon stored within in a very drastic way. Big uncertainties exist about the vulnerability of millenial old carbon to be mineralized and leave the thawing soils via CO2 or CH4. Thus the study aims for the better understanding of highly important topic, the long term fate of permafrost soil carbon. I have a strong major concern with the layout of the study. Using a space for time approach the authors compare three single soil pits with thousands of miles distance in between. The authors take the data of 3 soil pits and model soil OC development over 300 years into the future. All uncertainties, all vegetation and climate and parent material differences are just neglected, and the whole model is based on some 14C and C data. The results look feasible (of course there will be depth trends in OC), and they might be if you think a Gelisol might become an Inceptisol and Mollisol

with changing from a 300 mm to 800 mm precipitation ecosystem. But the whole study overstretches the space for time approach by far! It is already complicated to correlate soils in one catchment using this approach, but on completely different parent materials and ecosystems... The warming Arctic and its OC fate is a big topic, but is this worth putting together old data and squeezing it into a questionable modelling approach? You have a nice data set, so maybe its worth rethinking your approach and re-write it with what it is, three single soil pits. Based on that you could really go into detail discussing OM distribution and possibly also stabilization, but not telling a story that "this" Gelisol might be "this" Mollisol in 300 years. Please see detailled comments below: page 2 line 31 and following - If you only look into Hugelius this might be right for the permafrost regions, but there is a growing number of studies on subsoils globally. With this there is also a growing understanding of what drives subsoil C stabilization. There is also already some work on SOM fractions in the Arctic, so maybe worth checking for OM vulnerability in permafrost soils (e.g. Gentsch et al. EJSS 2015; Mueller et al GCB 2015). page 3 line 25-29 I doubt that todays arctic permafrost soils can via a space for time approach be related with Iowa soils. Space for time approaches even when conducted in the exact same ecosystem have a tremendous number of assumptions. In your case you are pushing these assumptions far of a meaningful level. page 3 line 30 and following - I clearly doubt that the research sites can give you a reliable answer. Of course you see differences between the sites, but what are the factors driving these changes, definitely not just a permafrost you find at one spot but not the other! page 4 line 8-10 Please give detailed mineralogy together with pedogenic oxides to show comparability of study sites with respect to aggregation and organo-minerl associations. page 4 line 14 - You get Tephra in at 1 m, so how are you dealing with different nutrient contents? page 5 line 4 - Actually approx. 3000 miles in between sites. page 7 line 8-17 On top of the differences you also compare soils with and without carbonates? Even under comparable climate you'll have differences in OC storage/stability due to carbonates vs. no-carbonates. page 7 line 18 - How representative were these single soil pits for the area (bulk density, mineralogy, C/N etc.), and thus how representative

to relate these soil types? page 7 line 18-22 What density agent was used? Please briefly describe the procedure. page 7 line 21 - What is your "occluded fraction", a light fraction or a mixed particulate together with minerals fraction? page 8 line 2-5 - On what was the OC input based, field data, assumptions? What are the input rates of the fractions? Wer differences in OM chemistry of the input taken into account? page 8 line 7-4 You are taking a modelling approach from a study that models physical OC transport at profile and landscape scale, to model depth functions of OC stability/mean residence time. The assumptions are based on soils from Iowa, but taken to the continuous permafrost Arctic. How are permafrost table depth, root input etc. related to your model assumptions? page 8 lines 15-20 - You are leaving out the unique features of permafrost soils by neglecting the vast amount of OC stored at depth. This also completely neglects soil erosion and changes in hydrologic conditions with permafrost thaw, which are well known to tremendously affect OC storage/fate/turnover. But those would be the step in between your studied sites. page 9 line 1-2 - This assumption is so far off! There are tremendous degrees of uncertainty already for concepts like "storage potential" but definitely for the fate of OC in permafrost soils. You are modelling your data to 200 cm depth, and obviously hit a cryoturbated pocket in the Gelisol in the 14C data. The other soils were sampled much shallower but you assume something underneath, which is definitely highyl speculative especially given the sight underlain by Tephra. With your approach you could also go a step further and include hot aride loess soils in central China. page 11 line 8-14 This is all based on assumptions! You did not measure a single k for any of the fractions. This might be vaque for one site, but for a comparison and especially as a base for forward modelling, this is far off! page 13 line 12 following - The whole paragraph is only based on assumptions! Where is rooting patterns and biomass data? Were is mineralogy/hydrology data? While the 14C data is based on soil horizons as well as depth layers, which kills comparability already. Okay, not to mention you bring up a story on sites thousands of miles away from each other... page 14 line 9-10 - You don't have any loss between the two. The only thing you have is three sites with different OC stocks, distribution and composition

and you model this data. So you could maybe "assume" differences in these measures between the analysed soils, but to relate them on a timescale of 300 years - this is not based on data! page 14 line 28 and following - This is all right, you demonstrated differences in the distribution of free vs. occluded POM and mineral-associated OM. But you can not draw a line between these distant soil types with respect to one develops from another.

---

## Referee Comment (RC2) · Anonymous Referee #2 · 28 Feb 2019

Harden et al. used the space for time approach in order to get insights on the critical question about the fate of permafrost soil carbon under climate change. The authors combined physico-chemical fractionation of soil C pools with radio carbon dating and exponential equation fitting with soil depth. The results showed depth distributions of organic C were related mainly to depths of rooting and changes in bulk density. According to the study, thawing of PF will cause changes in specific C pools. The first period until the year 2100 will result in net C loss of unprotected pools, while mineral protected pools will gain C. Further warming beyond 2100 will cause losses from the mineral protected C pools, while deeper rooting stimulate the gain of light fraction materials. These results are of strong importance for the scientific community. Not only for permafrost research, but also for general information about changes in

stabilization mechanisms of soil C under a future climate. The authors did a great job evaluating 14C in different SOM pools, which is crucial in order to understand SOM stabilization mechanisms. The study is written in excellent scientific English and well organized.

I found, however, some drawbacks which need to be considered further in the review. The authors provide only little information about parent material, except that there is some loess underlain. It would be good to have a map of the sites or some information about the depth of the loess sediments and the material below. Texture and mineralogical composition are crucial parameters for the storing capacity of OC in mineral soil layers. OC contents strongly correlate positively with mineral parameters such as clay, silt, Fe-Al- hydroxides in soil. Already small changes in these parameters have strong impact on the overall OC storage capacity. The space for time approach assumes that these parameters are similar between the sites. Unfortunately, no information on texture are presented. An increase of clay content by only 5% can result, for example, in up to 2% higher OC concentrations in temperate arable soils. If for example, the Inceptisols would have the highest clay content, than the gain in the mineral-associated fraction could be explained by that. Similar, the loss toward Mollisols could be explained by a slightly lower clay or Fe content. I'm afraid that the massage could be biases without considering these very important parameters. Incorporating these parameters in mixed effects models, or at least showing that clay is not a principle driver for OC stock change between the sites should solve the problem. Further, the gradient of sampling sited not only reflects a temperature gradient but also a precipitation gradient, from 270 mm in Gelisols to 850mm in Iova. How does the climate scenarios reflect changes in precipitation in the arctic? Precipitation and thus, soil moisture are next to temperature, the main drivers for OC mineralization. Therefore, it would be good to read how this moisture gradient reflects the model results.

General comments: I found no information's about how many profiles or soil samples have been analyses. Also the data in supplement were not very helpful. How many

samples have been fractionated?

Specific comments: P2L3: "Fitting an exponential equation to depth trends in soil C..." please explain if specific pools are fitted or the bulk soil. The same for the depths of rooting and changes in bulk density. Pools or bulk? P3L3-12: the paragraph described that SOC stocks and MRT depend from environmental and substrate-specific factors. In terms of substrates, the authors refer mainly to the quality and quantity of plant residue inputs. One crucial factor for the SOC storing quantity is the parent material or the substrate for soil formation. Clay, silt and Fe-Al-(oxy)hydroxide content effecting the overall storage capacity of SOC (Kleber et al., 2015; von Lützow et al., 2006). This should be mentioned here, because mineral-organic interactions are part of the manuscript. There are also some latest works on organo-mineral stabilization in permafrost soils (Gentsch et al., 2015, 2018; Mueller et al., 2017). P4L16 following: Please describe how the samples were taken. How was bulk density measured, which is used in calculating the C density? P5L12: "dramatic differences" sounds a bit fishy. Please chance the phrase. P8L15: I found it pretty hard to understand what Zmin means. Would be nice to have a quick excess explanation P9L14: please delete relatively before modern. Everything F>1 is per definition modern. So the whole profile LF is modern. P12-13 Fig 4: there is probably a mistake in units by description of the model results from Fig 4. In the text the changes are given in g C m-2 which is reasonable for me. Figure 4 reported values in kg C m-2 y-1 which resulted in incredible amounts of C when scaling them up to a larger area or over 200years. P13L17: correct Zmin lower case. Also in later sentences. Figure 3: please specify how many profiles were involved in the model

References: Gentsch, N., Mikutta, R., Alves, R. J. E., Barta, J., Čapek, P., Gittel, A., Hugelius, G., Kuhry, P., Lashchinskiy, N., Palmtag, J., Richter, A., Šantrůčková, H., Schnecker, J., Shibistova, O., Urich, T., Wild, B. and Guggenberger, G.: Storage and transformation of organic matter fractions in cryoturbated permafrost soils across the Siberian Arctic, Biogeosciences, 12(14), 4525–4542, doi:10.5194/bg-12-4525-2015,

2015.

Gentsch, N., Wild, B., Mikutta, R., Čapek, P., Diáková, K., Schrumpf, M., Turner, S., Minnich, C., Schaarschmidt, F., Shibistova, O., Schnecker, J., Urich, T., Gittel, A., Šantrůčková, H., Bárta, J., Lashchinskiy, N., Fuß, R., Richter, A. and Guggenberger, G.: Temperature response of permafrost soil carbon is attenuated by mineral protection, Glob. Change Biol., 24(8), 3401–3415, doi:10.1111/gcb.14316, 2018.

Kleber, M., Eusterhues, K., Keiluweit, M., Mikutta, C., Mikutta, R. and Nico, P. S.: Mineral–Organic Associations: Formation, Properties, and Relevance in Soil Environments, Adv. Agron., 130, 1–140, 2015.

von Lützow, M. v., Kögel-Knabner, I., Ekschmitt, K., Matzner, E., Guggenberger, G., Marschner, B. and Flessa, H.: Stabilization of organic matter in temperate soils: mechanisms and their relevance under different soil conditions – a review, Eur. J. Soil Sci., 57(4), 426–445, doi:10.1111/j.1365-2389.2006.00809.x, 2006.

Mueller, C. W., Hoeschen, C., Steffens, M., Buddenbaum, H., Hinkel, K., Bockheim, J. G. and Kao-Kniffin, J.: Microscale soil structures foster organic matter stabilization in permafrost soils, Geoderma, 293, 44–53, doi:10.1016/j.geoderma.2017.01.028, 2017.

---

## Referee Comment (RC3) · Anonymous Referee #3 · 1 Mar 2019

The present manuscript entitled "Beneath the arctic greening: Will soils lose or gain carbon or perhaps a little of both?" deals with potential responses of soil systems following ecosystem shifts related to climate change. The authors used a space-for-time sequence of soils developed on loess substrates along a cold-to-warm climatic gradient (from soils with permafrost to soils lacking permafrost) with temperatures representative for Interior Alaska in the years 2014, 2100 and 2300. Soil profiles of 2m depth were sampled and analyzed for organic C in bulk soil as well as in density fractions (free light, occluded light and mineral-associated fraction) and 14C data. Using the collected data and modeling, the authors aim at predicting the fate of soil C upon the arctic warming. The major findings are that (i) "depth distributions of organic C were related mainly to depths of rooting and changes in bulk density", (ii) "thawing from the Gelisol to Inceptisol in loess parent materials from present to year 2100 resulted in small net gains to soil C, reflecting the net balance between loss of detrital and gain into occluded and mineral associated C" and (iii) "greater warming and shifts from Inceptisol to Mollisol analogous to predicted warming from circa 2100 to 2300 resulted in net losses from both occluded and mineral associated C...". The response of C stabilization upon climate change and especially the fate of soil C upon thawing of permafrost is a highly relevant, up to date topic, which will be of interest to a broad readership and it fits into the scope of "SOIL". I personally find the idea underlying this paper nice and the results interesting. However, I have some serious concerns about the used space for time approach – although the authors took care about the parent material (loess deposits) and kept it similar for all three soils investigated there are many assumptions underlying the space-for-time approach that are not discussed or even mentioned at all. In the following, I will list my major concerns / the shortcomings of the approach used which need to be clearly addressed and discussed before the manuscript can be considered for publication in SOIL.

General comments (1) The authors do not provide any data on the parent material except for the fact that it is "loess deposits originating from [...] the Matinuska and Knik glaciers (Inceptisols), glacial to post-glacial outwash along the Missouri River and distal loess sources in Nebraska (Mollisols)". The source of the parent material for the Gelisol is not even mentioned at all. Please provide additional information on the soil profiles investigated: (a) where are the profiles located? How far are profiles from each of the three soil types investigated apart from each other? I suggest adding a map of all soil profiles investigated. (b) What about the mineralogical composition of the parent material of each soil profile? What is the texture? Are Fe-/ Al- oxides or hydroxides present? Are carbonates present? All these parameters strongly influence soil C concentrations, amounts of mineral-associated and particulate C as well as C stability in soil. Please give additional information in your table if available. If not available, these factors should at least be clearly taken account of and be discussed! (c) How sure are the authors about the exact development of their soils, i.e. how sure

is the assumption that a Gelisol under Alaskan Black Spruce forest will develop to a Mollisol under a grassland ecosystem? Is it possible that other vegetations/ecosystems would develop (e.g. mixed or deciduous forests) and what would that mean in terms of the C distribution and stability in the soil? (2) The conclusions drawn from the data are a bit too far stretched or let's say formulated too general. E.g. p. 2, l. 8-10 "Thawing from Gelisol to Inceptisol in loess parent materials from present to year 2100 resulted in small net gains to soil C, reflecting..." or the sentence thereafter (p. 2, l. 10-12 "Greater warming and shifts from Inceptisol to Mollisol analogous to predicted warming from circa 2100 to 2300 resulted in net losses from both occluded and mineral associated C" – the authors did not observe / measure C gains or losses during soil development, they only PREDICT these upon ASSUMED / PREDICTED soil development during global warming. Therefore, you only ASSUME that this happens. Please phrase your conclusions more carefully.

Minor comments and editorial comments - p. 2, l. 1: The authors write that all profiles developed on similar geologic substrate, i.e. "wind-blown loess deposits". However, on p.5, l. 5-6 they write that Mollisols were formed on "loess deposits originated from glacial to post-glacial outwash along the Missouri River and distal loess sources..." - p. 4 "2.1 Study sites": Information on how many profiles per soil type were collected are missing. One can draw the information from the tables, but this is cumbersome – please add here Further, it looks to me that the number of profiles collected differs between soil types (4 profiles for Gelisols, 3 profiles for Mollisols and 14 for Inceptisols – is that correct?) – how was this taken into account statistically / in modelling? - P. 7, l. 18-23: Please provide a bit more information on the density fractionation: (a) what density agent was used? In case it was sodium polytungstate or another salt (b) how did the authors get rid of salt remaining in the sample? Washing? - p. 8, l. 14: different font - p.9, l.8: I suggest writing "The C concentration (C%) decreased with..." instead of "C percent decreased with..." - p. 9, l. 9: change Molliols to Mollisols - p. 9, l. 8-10: "Parameters of $Z^*$ are deeper for Gelisols and Inceptisols than for Mollisols , whereas $Z_{min}$ parameters are shallower for Gelisols and Inceptisols than Mollisols." First, what

exactly does this mean? A deeper $Z_{min}$ means that the lowest C concentration was reached in a deeper soil depth? What exactly means a "deeper parameter of $Z^*$"? Second, if I read Table 1 correctly, $Z^*$ for Gelisols range from 14.8 to 33.8, $Z^*$ for Inceptisols range from 9.8 to 64.3 and $Z^*$ for Mollisols range from 35.5 to 56.6. So I would not underline that "parameters for $Z^*$ are deeper for Gelisols and Inceptisols than Mollisols"? The same holds true for $Z_{min}$... Or did I get something wrong here? - p. 13, l. 17: Manuscripts in preparation, i.e. non-accepted, should not be cited. - p. 14, l. 21: write "C" instead of carbon - p. 14, l. 22: delete one dot after "...spruce.." - p. 14, l. 23: =0.87 instead of .87 - p. 15, l.23: write "C" instead of carbon - p. 15, l. 26-28: "our space-for-time approach integrates changes in vegetation, climate, and mineral factors to provide..." What mineral factors are integrated? Where is the data for that? - Table 1: (A) I would change the order of the soil types from "Gelisol – Mollisol – Inceptisol" to the order of the assumed soil development, i.e. "Gelisol – Inceptisol – Mollisol" (B) I had to search for the meaning of each parameter in the Material and Methods section and this is cumbersome. Please add a short explanation of all parameters listed in Table 1 either in the header or as a footnote (e.g. $C_s$ – surface C, $C_{min}$ – minimum C etc.) (C) Additionally, I did not even find an explanation for all of the parameters: What are the adjusted parameters ($Z_{adj}$, $Z^*_{adj}$, $Z_{min\_adj}$?) – what did you do to adjust them? This information needs to be added in the Materials and Methods section. (D) Please add units for the parameters given

- Table 2: Please add a column with the soil type here as you did in table 1 - Table S1: Profile HCCN2/3 starts at 24 cm soil depth? What happened to the upper 23 cm?

---

## Author Comment (AC1) · 1 Mar 2019

Please find below our responses to the general comments from Referee 1.

*General Comment 1. I have a strong major concern with the layout of the study. Using a space for time approach the authors compare three single soil pits with thousands of miles distance in between. The authors take the data of 3 soil pits and model soil OC development over 300 years into the future. All uncertainties, all vegetation and climate and parent material differences are just neglected, and the whole model is based on some 14C and C data. The results look feasible (of course there will be depth trends in OC), and they might be if you think a Gelisol might become an Inceptisol and Mollisol C1 SOILD Interactive comment Printer-friendly version Discussion paper with*

*changing from a 300 mm to 800 mm precipitation ecosystem. But the whole study overstretches the space for time approach by far! It is already complicated to correlate soils in one catchment using this approach, but on completely different parent materials and ecosystems...*

We appreciate Referee 1's concerns about our application of the space-for-time substitution approach to this study. However we disagree with the assertion that state factors are "are just neglected." As we state in the manuscript, we aimed to control for parent material across the chronosequence by only sampling late Pleistocene loess soils. While these three soils fall under different taxa, they are off similar origin. We also conducted particle size analysis on all three soil profiles, which we will include in revisions at the recommendation of Referee 2. These data further support our consideration of parent materials across the chronoequence.

Second, we accounted for relief across the chronosequence, which is another state factor described by Jenny (1941). All three soil profiles were sampled on hillslopes. We will add text to the Methods to better describe topographic position of sampling sites.

Third, we acknowledge in the Methods that vegetation and climate have varied across the chronosequence. Figure 1 shows the close relationship between climate (i.e., soil temperature) and time. More specifically, present-day soil temperatures across the chronosequence closely track projected changes in soil temperature for a permafrost site out to 2100 and 2300. While we were not able to control for vegetation, this is a common issue with soil chronosequence studies, including upland fire chronosequences (O'Donnell at al., 2011), peatland thaw chronosequences (O'Donnell et al. 2012), and deglaciation and especially peatland chronosequences (Trumbore et al, 1997). In our next draft we will add text to better support our approach, identify our assumptions, and highlight possible limitations.

*General Comment 2. The warming Arctic and its OC fate is a big topic, but is this worth*

*putting together old data and squeezing it into a questionable modelling approach? You have a nice data set, so maybe its worth rethinking your approach and re-write it with what it is, three single soil pits. Based on that you could really go into detail discussing OM distribution and possibly also stabilization, but not telling a story that "this" Gelisol might be "this" Mollisol in 300 years.*

We would prefer to maintain the model in the manuscript for the following reasons.

First, a primary objective of the manuscript was to illustrate one possible application of the data through a very simple modeling framework. Through this simple modeling approach, we were able to highlight dynamics of different soil fractions given variable 14C-based turnover estimates in response to ecosystem changes. In many modeling studies, turnover rates for different soil pools are derived empirically from incubation of bulk soils or ecosystem fluxes, not from observations of different pools. Thus, our study represents a novel approach, including both observational and modeling approaches for specific soil C fractions.

Second, another primary goal of our study was to provide some constraints on possible C changes following thawing of ice-rich Pleistocene loess. This is a globally important C pool, and the fate of this C is poorly constrained both by observations and models, particularly at the decadal and century time scale. Other data-driven modeling approaches have used relatively short-term incubation data (month to annual time scale) to drive decomposition rates to estimate the permafrost-carbon feedback (e.g., Koven et al. 2015). Our radiocarbon-based technique is a more appropriate approach for constraining C dynamics over longer time scales.

Third, outcomes of the modeling work should not be interpreted literally (i.e., a Gelisol might become a Mollisol in 300 years). While our simple model is based on observations, results should be interpreted with caution, given the limitations of our approach. The goal of the model was not necessarily to be predictive, but to better understand dynamics associated with ecosystem change. We will add text to the Discussion to

better articulate what we should AND should not conclude from our modeling results.

---

## Author Comment (AC2) · 5 Apr 2019

Response to Anonymous Referee #1 General Comments

1. I have a strong major concern with the layout of the study. Using a space for time approach the authors compare three single soil pits with thousands of miles distance in between. The authors take the data of 3 soil pits and model soil OC development over 300 years into the future. All uncertainties, all vegetation and climate and parent material differences are just neglected, and the whole model is based on some 14C and C data. The results look feasible (of course there will be depth trends in OC), and they might be if you think a Gelisol might become an Inceptisol and Mollisol with changing from a 300 mm to 800 mm precipitation ecosystem. But the whole study

overstretches the space for time approach by far! It is already complicated to correlate soils in one catchment using this approach, but on completely different parent materials and ecosystems...

We appreciate Referee #1's concerns about our application of the space-for-time substitution approach to this study. However we disagree with the assertion that state factors are "are just neglected." As we state in the manuscript and now clarify further, we aimed to control for parent material, age, and slope across the study sites. We have revised the text in section 2.1 as follows:

"Soil samples were collected and analyzed from three study sites including a Gelisol in interior Alaska (n = 4 profiles), an Inceptisol in south-central Alaska (n = 14), and a Mollisol in Iowa (n = 3). Parent material, time (age), and topography are three of the primary soil-forming factors known to influence soil properties (Jenny 1941) and soils for this study were similarly underlain by late Pleistocene loess (silty sediment of wind-blown origin) as a common age and type of parent material and slopes of 3-10% as a common topographic setting. Other soil forming factors (climate, vegetation) varied across sites, providing a means to test the effects of changing climate and ecosystem state on soil C storage and flux. For example, Iowa Mollisols formed with little or no permafrost throughout most of the depositional history of loess accumulation and organic matter burial. Alaska Inceptisols formed with no permafrost since at least 5,000 y BP as evidenced by a 5 ka volcanic tephra at ∼1 m depth that was not deformed by frost heave; and Alaska Gelisols formed with continuous permafrost since the Pleistocene."

We now have included information on soil texture and particle size in the Methods section, and have provided a link to the data at the International Soil Radiocarbon Database. These data further support our consideration of parent materials across the chronoequence. Also see section 2.1 for each Gelisol, Inceptisol and Mollisol. (clay content ranges within and between sites from about 5 to 20% clay).

As described above, we accounted for relief across the chronosequence, which is another state factor described by Jenny (1941). All three soil profiles were sampled on hillslopes of 1-10% slope. We added text to the Methods to better describe topographic position of sampling sites.

We acknowledge and clarify in the Methods that vegetation and climate have varied across the sequence. Figure 2 (formerly Figure 1) shows the close relationship between climate (i.e., soil temperature) and time. More specifically, present-day soil temperatures across the sequence closely track projected changes in soil temperature for a permafrost site out to 2100 and 2300. To be clear this study offers a constraining conceptual model in which climate, ecosystem and soil state are represented by current day steady-state systems. We tried to not overstate this conceptual model but feel strongly that the science community could use current climate-ecosystem-soil associations to constrain (in particular) dynamic vegetation models. We have added and clarified text throughout the manuscript to better support our approach, identify our assumptions, and highlight possible limitations. For example on page 4, we refer to our "space-for-time substitution approach" and that "We modeled the gradient as a warming scenario in order to conceptualize how climate, ecosystem shift, and soil state might transition from their steady state"

2. The warming Arctic and its OC fate is a big topic, but is this worth putting together old data and squeezing it into a questionable modelling approach? You have a nice data set, so maybe its worth rethinking your approach and re-write it with what it is, three single soil pits. Based on that you could really go into detail discussing OM distribution and possibly also stabilization, but not telling a story that "this" Gelisol might be "this" Mollisol in 300 years.

We choose to maintain this exercise for the following reasons. First, a primary objective of the manuscript was to illustrate one possible application of the data through a very simple modeling framework. Through this simple modeling approach, we were able to highlight dynamics of different soil fractions given variable 14C-based turnover estimates in response to ecosystem changes. In many modeling studies, turnover rates

for different soil pools are derived empirically from incubation of bulk soils or ecosystem fluxes, not from observations of different pools. Thus, our study represents a novel approach, including both observational and modeling approaches for specific soil C fractions.

Second, another primary goal of our study was to provide some constraints on possible C changes following thawing of ice-rich Pleistocene loess. This is a globally important C pool, and the fate of this C is poorly constrained both by observations and models, particularly at the decadal and century time scale. Other data-driven modeling approaches have used relatively short-term incubation data (month to annual time scale) to drive decomposition rates to estimate the permafrost-carbon feedback. Our radiocarbon-based technique is a more appropriate approach for constraining C dynamics over longer time scales.

Third, while ecosystem transitions are commonly modeled based on climate shifts (both for back-casting and forecasting) the link between aboveground (vegetation) and belowground (soils) generally are based on biogeochemistry models and not on ecosystem – soil associations. This rather deliberate approach to ecosystem-soil opens up entirely new data from soil surveys associated with land cover and land use change that we think can strengthen our understanding of these linkages.

Last, outcomes of the modeling work should not be interpreted literally (i.e., a Gelisol might become a Mollisol in 300 years). While our simple model is based on observations, results should be interpreted with caution, given the limitations of our approach. The goal of the model was not necessarily to be predictive, but to better understand dynamics associated with ecosystem change. We also added a new conceptual diagram (Figure 1) at the request of the Topical Editor.

Specific Comments 1. page 2 line 31 and following: If you only look into Hugelius this might be right for the permafrost regions, but there is a growing number of studies on subsoils globally. With this there is also a growing understanding of what drives subsoil

C stabilization. There is also already some work on SOM fractions in the Arctic, so maybe worth checking for OM vulnerability in permafrost soils (e.g. Gentsch et al. EJSS 2015; Mueller et al GCB 2015).

We added text here to clarify that we are specifically referring to soils "in the northern permafrost region." We also added as sentence to reference to work of Mueller et al. (2015).

2. page 3 line 25-29 I doubt that todays arctic permafrost soils can via a space for time approach be related with Iowa soils. Space for time approaches even when conducted in the exact same ecosystem have a tremendous number of assumptions. In your case you are pushing these assumptions far of a meaningful level. page 3 line 30 and following - I clearly doubt that the research sites can give you a reliable answer. Of course you see differences between the sites, but what are the factors driving these changes, definitely not just a permafrost you find at one spot but not the other!

Without overstating our "conceptual experiment", if there indeed are new grass-dominated ecosystems in a drastically warmer arctic, it is conceivable that Mollisols will form underneath them and that their roots will colonize deep, unfrozen substrate. After all, there are some Mollisols in Alaska today. Moreover, we emphasize that the important differentiation among the sites is the association of climate-ecosystem-soil and that indeed the soil carbon dynamics in each site are indeed indicative of carbon storage and turnover for those associations. On this point we will have to disagree. Please see how we handled your objection by reviewing the discussion section, in particular on pages 15-16: "...these comparisons illuminate the potential for these (physical, chemical, biological) mechanisms to shift under changing climatic and ecosystem states." Also, "Several important caveats should be noted in this approach", and "Our approach is a comparison e.g. a space-for-time/climate/ecosystem shift and is not literally a process-based model but rather is an exercise to more deeply understand the how and why soil carbon might be stabilized or destabilized in differing climate and ecosystem states.

2. page 4 line 8-10 Please give detailed mineralogy together with pedogenic oxides to show comparability of study sites with respect to aggregation and organo-mineral associations.

Citations within the manuscript provide detailed field and laboratory data, as do the databases in which those data now reside (International Soil Radiocarbon Database)

3. page 4 line 14 - You get Tephra in at 1 m, so how are you dealing with different nutrient contents?

We have no explicit form of nutrient associations with our carbon data.

4. page 5 line 4 - Actually approx. 3000 miles in between sites.

Yes, this is true.

5. page 7 line 8-17 On top of the differences you also compare soils with and without carbonates? Even under comparable climate you0 ll have differences in OC storage/stability due to carbonates vs. no-carbonates.

All of our data are carbonate free and indeed there was no evidence for carbonate in these soils.

6. page 7 line 18 - How representative were these single soil pits for the area (bulk density, mineralogy, C/N etc.), and thus how representative to relate these soil types?

While other data exist and we encourage more elaborate modeling with regard to new ecosystem-soil associations in the future. This paper is not tackling spatial variation across soil profiles and sites.

7. page 7 line 18-22 What density agent was used? Please briefly describe the procedure.

We added text to note that "sodium polytungstate" was the density solution, and we describe the procedure in depth on Page 7.

8. page 7 line 21 - What is your "occluded fraction", a light fraction or a mixed particulate together with minerals fraction?

We added extensive text on page 7 to clarify how each fraction was determined and defined, including the "occluded" fraction.

9. page 8 line 2-5 - On what was the OC input based, field data, assumptions? What are the input rates of the fractions? Were differences in OM chemistry of the input taken into account?

Input is not explicitly modeled in this paper.

10. page 8 line 7-4 You are taking a modelling approach from a study that models physical OC transport at profile and landscape scale, to model depth functions of OC stability/mean residence time. The assumptions are based on soils from Iowa, but taken to the continuous permafrost Arctic. How are permafrost table depth, root input etc. related to your model assumptions?

This is not a dynamic model, rather it is a calculation from one steady-state system to a new steady-state system. The simplicity of this approach is its strength – if today's ecosystem-soil association is taken as a hypothetical proxy and if we know something about tomorrow's distribution of ecosystems, then we simply calculate how the soil carbon might change along with the ecosystem. This transient response along the way to the new state – such as changes in permafrost table, vegetation are not specified. We've added text to the manuscript to clarify this point in numerous spots.

11. page 8 lines 15-20 - You are leaving out the unique features of permafrost soils by neglecting the vast amount of OC stored at depth. This also completely neglects soil erosion and changes in hydrologic conditions with permafrost thaw, which are well known to tremendously affect OC storage/fate/turnover. But those would be the step in between your studied sites.

Correct. While there are many unique processes specific to thaw in permafrost terrain, we are simply considering transition from one steady-state ecosystem to the next, without accounting for a specific mechanism of change (e.g., erosion).

12. page 9 line 1-2 - This assumption is so far off! There are tremendous degrees of uncertainty already for concepts like "storage potential" but definitely for the fate of OC in permafrost soils. You are modelling your data to 200 cm depth, and obviously hit a cryoturbated pocket in the Gelisol in the 14C data. The other soils were sampled much shallower but you assume something underneath, which is definitely highly speculative especially given the sight underlain by Tephra. With your approach you could also go a step further and include hot aride loess soils in central China.

We have added text throughout the manuscript to better describe the assumptions and limitations of our approach.

13. page 11 line 8-14 This is all based on assumptions! You did not measure a single k for any of the fractions. This might be vaque for one site, but for a comparison and especially as a base for forward modelling, this is far off!

Decomposition coefficients (i.e., k values) are based on turnover time constrained by radiocarbon in each fraction using a steady-state modeling approach (e.g., Trumbore 1993).

14. page 13 line 12 following - The whole paragraph is only based on assumptions! Where is rooting patterns and biomass data? Were is mineralogy/hydrology data? While the 14C data is based on soil horizons as well as depth layers, which kills comparability already. Okay, not to mention you bring up a story on sites thousands of miles away from each other...

Rooting data are in Table S1 as field descriptions of abundance and size. There are no biomass data, and mineralogy data are in citations. Our 14C data are for depth and fraction – we are uncertain what the reviewer is referring in their critique. See above for revised text.

15. page 14 line 9-10 - You don't have any loss between the two. The only thing you have is three sites with different OC stocks, distribution and composition and you model this data. So you could maybe "assume" differences in these measures between the analysed soils, but to relate them on a timescale of 300 years - this is not based on data! page 14 line 28 and following - This is all right, you demonstrated differences in the distribution of free vs. occluded POM and mineral-associated OM. But you can not draw a line between these distant soil types with respect to one develops from another.

We changed wording to "comparison and postulated transition. . .alludes to losses". We have taken steps throughout the manuscript to remind readers that this is a comparison set up as a conceptual "space-for-time" model.

---

## Author Comment (AC3) · 5 Apr 2019

Response to Anonymous Referee #2 General Comments

1. Harden et al. used the space for time approach in order to get insights on the critical question about the fate of permafrost soil carbon under climate change. The authors combined physico-chemical fractionation of soil C pools with radio carbon dating and exponential equation fitting with soil depth. The results showed depth distributions of organic C were related mainly to depths of rooting and changes in bulk density. According to the study, thawing of PF will cause changes in specific C pools. The first period until the year 2100 will result in net C loss of unprotected pools, while mineral protected pools will gain C. Further warming beyond 2100 will cause losses

from the mineral protected C pools, while deeper rooting stimulate the gain of light fraction materials. These results are of strong importance for the scientific community. Not only for permafrost research, but also for general information about changes in stabilization mechanisms of soil C under a future climate. The authors did a great job evaluating 14C in different SOM pools, which is crucial in order to understand SOM stabilization mechanisms. The study is written in excellent scientific English and well organized.

We appreciated the positive feedback.

2. I found, however, some drawbacks, which need to be considered further in the review. The authors provide only little information about parent material, except that there is some loess underlain. It would be good to have a map of the sites or some information about the depth of the loess sediments and the material below. Texture and mineralogical composition are crucial parameters for the storing capacity of OC in mineral soil layers. OC contents strongly correlate positively with mineral parameters such as clay, silt, Fe-Al- hydroxides in soil. Already small changes in these parameters have strong impact on the overall OC storage capacity. The space for time approach assumes that these parameters are similar between the sites. Unfortunately, no information on texture are presented. An increase of clay content by only 5% can result, for example, in up to 2% higher OC concentrations in temperate arable soils. If for example, the Inceptisols would have the highest clay content, than the gain in the mineral-associated fraction could be explained by that. Similar, the loss toward Mollisols could be explained by a slightly lower clay or Fe content. I'm afraid that the massage could be biases without considering these very important parameters. Incorporating these parameters in mixed effects models, or at least showing that clay is not a principle driver for OC stock change between the sites should solve the problem. Further, the gradient of sampling sited not only reflects a temperature gradient but also a precipitation gradient, from 270 mm in Gelisols to 850mm in Iova. How does the climate scenarios reflect changes in precipitation in the arctic? Precipitation and thus, soil moisture are next to

temperature, the main drivers for OC mineralization. Therefore, it would be good to read how this moisture gradient reflects the model results.

We included more information about particle size and citations to mineralogy AND we entered those data into the online repository for these profiles (International Soil Radiocarbon Database). It is true that the parent materials are not exactly comparable among sites (e.g., clay contents, Fe oxides, etc vary) but their variation is likely far less than that of climate and biotic systems which is the basis for the comparison. As for precipitation, we don't know. We included this "unknown" in the caveats paragraph on page 16.

3. I found no information's about how many profiles or soil samples have been analyses. Also the data in supplement were not very helpful. How many samples have been fractionated?

We have added text to show the number of profiles analyzed in the Methods section. We also note that only one profile of soil samples were fractioned for each soil type.

Specific Comments

1. Specific comments: P2L3: "Fitting an exponential equation to depth trends in soil C: : :" please explain if specific pools are fitted or the bulk soil. The same for the depths of rooting and changes in bulk density. Pools or bulk?

While we fit the exponential equation to both bulk and fractionated soils, this statement specifically refers to depth trends and controls on bulk soil samples. We revised text in the Abstract to reflect this.

2. P3L3-12: the paragraph described that SOC stocks and MRT depend from environmental and substrate-specific factors. In terms of substrates, the authors refer mainly to the quality and quantity of plant residue inputs. One crucial factor for the SOC storing quantity is the parent material or the substrate for soil formation. Clay, silt and Fe-Al-(oxy)hydroxide content effecting the overall storage capacity of SOC (Kleber et

al., 2015; von Lützow et al.,2006). This should be mentioned here, because mineral-organic interactions are part of the manuscript. There are also some latest works on organo-mineral stabilization in permafrost soils (Gentsch et al., 2015, 2018; Mueller et al., 2017).

Excellent point. We added text and some citations recommended by the reviewer to this paragraph.

3. P4L16 following: Please describe how the samples were taken. How was bulk density measured, which is used in calculating the C density?

We added text to this section of the Methods to this section to better describe how soils were sampled and bulk density was determined.

4. P5L12: "dramatic differences" sounds a bit fishy. Please chance the phrase.

We changed the wording here to state "considerable differences".

5. P8L15: I found it pretty hard to understand what Zmin means. Would be nice to have a quick excess explanation

We added a sentence here to better define Zmin and also Cmin.

6. P9L14: please delete relatively before modern. Everything F>1 is per definition modern. So the whole profile LF is modern.

Agreed. We deleted "relatively" from this sentence.

7. P12-13 Fig 4: there is probably a mistake in units by description of the model results from Fig 4. In the text the changes are given in g C m-2 which is reasonable for me. Figure 4 reported values in kg C m-2 y-1, which resulted in incredible amounts of C when scaling them up to a larger area or over 200years.

We revised the Results text in this section to better describe fluxes (and not stocks) in Figure 4. Based on these flux estimates, which are constrained by our radiocarbon

measurements, we provide total stock changes over relevant time scales.

8. P13L17: correct Zmin lower case. Also in later sentences.

We corrected the formatting of Zmin throughout the manuscript.

9. Figure 3: please specify how many profiles were involved in the model

We added text to the Figure legend to note that model-data fits are based on fractionated soils for one profile from each soil type.

---

## Author Comment (AC4) · 5 Apr 2019

Response to Anonymous Referee #3 General Comments 1. The authors do not provide any data on the parent material except for the fact that it is "loess deposits originating from [: : :] the Matinuska and Knik glaciers (Inceptisols), glacial to post-glacial outwash along the Missouri River and distal loess sources in Nebraska (Mollisols)". The source of the parent material for the Gelisol is not even mentioned at all. Please provide additional information on the soil profiles investigated:

(a) where are the profiles located? How far are profiles from each of the three soil types investigated apart from each other? I suggest adding a map of all soil profiles investigated.

[Figure]

In the "Section 2.1 Study Sites," we discuss the location of sites and provide GPS coordinates for each sampling locations, and we also refer to citations of original studies that generated these soil samples.

(b) What about the mineralogical composition of the parent material of each soil profile? What is the texture? Are Fe-/ Al- oxides or hydroxides present? Are carbonates present? All these parameters strongly influence soil C concentrations, amounts of mineral-associated and particulate C as well as C stability in soil. Please give additional information in your table if available. If not available, these factors should at least be clearly taken account of and be discussed!

All of our data are carbonate free and indeed there was no evidence for carbonate in these soils (pH acidic to neutral; no white precipitates). Textures are mostly silt loams but some silts and silt clay loams are present (see Suppl Table ST1) –there are no obvious correlations in texture-%C. We don't have consistent oxide data although such data may be added eventually to the repository at International Soil Radiocarbon Database.

(c) How sure are the authors about the exact development of their soils, i.e. how sure is the assumption that a Gelisol under Alaskan Black Spruce forest will develop to a Mollisol under a grassland ecosystem? Is it possible that other vegetations/ecosystems would develop (e.g. mixed or deciduous forests) and what would that mean in terms of the C distribution and stability in the soil?

"How sure?" Not at all. It would be fun to run an exercise based on probability functions for ecosystem types /soil types in various soil temperature space. If however there indeed are new grass-dominated ecosystems in a drastically warmer arctic, it is conceivable that Mollisols will form underneath them and that their roots will colonize deep, unfrozen substrate.

2. The conclusions drawn from the data are a bit too far stretched or let's say formulated too general. E.g. p. 2, l. 8-10 "Thawing from Gelisol to Inceptisol in loess parent

materials from present to year 2100 resulted in small net gains to soil C, reflecting: : :"
or the sentence thereafter (p. 2, l. 10-12 "Greater warming and shifts from Inceptisol to
Mollisol analogous to predicted warming from circa 2100 to 2300 resulted in net losses
from both occluded and mineral associated C" – the authors did not observe / measure
C gains or losses during soil development, they only PREDICT these upon ASSUMED
/ PREDICTED soil development during global warming. Therefore, you only ASSUME
that this happens. Please phrase your conclusions more carefully.

We made a number of edits to the Abstract and Discussion sections to better character-
ize our "exercise" so that it is not misinterpreted. Examples include: "The comparison
and postulated transition.." "alludes to"; "comparisons among. . .states" . "Model output
based on these comparisons suggests that. . ." are indicated by this comparison"

Specific Comments

1. p. 2, l. 1: The authors write that all profiles developed on similar geologic substrate,
i.e. "wind-blown loess deposits". However, on p.5, l. 5-6 they write that Mollisols were
formed on "loess deposits originated from glacial to post-glacial outwash along the
Missouri River and distal loess sources: : :" –

We omitted the text about "similar geologic substrates" and simply state that all profiles
developed from "wind-blown loess deposits."

2. p. 4 "2.1 Study sites": Information on how many profiles per soil type were collected
are missing. One can draw the information from the tables, but this is cumbersome
– please add here Further, it looks to me that the number of profiles collected differs
between soil types (4 profiles for Gelisols, 3 profiles for Mollisols and 14 for Inceptisols
– is that correct?) – how was this taken into account statistically / in modelling?

We added text in parentheses here to note the number of profiles per soil type. Sample
size did not impact depth fits, as Equation (1) was fit to each individual profile. Modeling
output was based on only one soil profile of fractionated soil per soil type.

3. P. 7, l. 18-23: Please provide a bit more information on the density fractionation: (a) what density agent was used? In case it was sodium polytungstate or another salt (b) how did the authors get rid of salt remaining in the sample? Washing?

We added several sentences here to clarify the methods for density fractionation.

4. p. 8, l. 14: different font

We corrected the font in this section.

5. p.9, l.8: I suggest writing "The C concentration (C%) decreased with: : :" instead of "C percent decreased with: : :"

We edited the text here following this recommendation.

6. p. 9, l. 9: change Molliols to Mollisols

We corrected the spelling of "Mollisols" in this sentence.

7. p. 9, l. 8-10: "Parameters of Z* are deeper for Gelisols and Inceptisols than for Mollisols , whereas Zmin parameters are shallower for Gelisols and Inceptisols than Mollisols." First, what exactly does this mean? A deeper Zmin means that the lowest C concentration was reached in a deeper soil depth? What exactly means a "deeper parameter of Z*"? Second, if I read Table 1 correctly, Z* for Gelisols range from 14.8 to 33.8, Z* for Inceptisols range from 9.8 to 64.3 and Z* for Mollisols range from 35.5 to 56.6. So I would not underline that "parameters for Z* are deeper for Gelisols and Inceptisols than Mollisols"? The same holds true for Zmin: : : Or did I get something wrong here?

We revised the text in this section as follows: "C concentration (C%) of bulk soil declines exponentially with depth, as captured by the depth parameters for Eq. 1 (Table 1). Values for Z* indicate an e-folding depth and are shallower for Gelisols (mean 24) and Inceptisols (27) than for Mollisols (43), whereas values for Zmin indicate the biotic zone and are deepest for Mollisols (120 compared to 57, 66 for Gelisol and Inceptisols).

Cdeep values are greatest for Gelisols (mean 1.1%C), followed by Inceptisols (0.9%) and Mollisols (0.1%). " 8. p. 13, l. 17: Manuscripts in preparation, i.e. non-accepted, should not be cited.

We omitted this citation from the text.

9. p. 14, l. 21: write "C" instead of carbon - p. 14, l. 22: delete one dot after ": : :spruce.."

We changed the text to "C' instead of "carbon" throughout the manuscript. We also deleted the extra period as indicated by the reviewer.

10. p. 14, l. 23: =0.87 instead of .87 - p. 15, l.23: write "C" instead of carbon - p. 15, l. 26-28: "our space-for-time approach integrates changes in vegetation, climate, and mineral factors to provide: : :" What mineral factors are integrated? Where is the data for that?

We corrected the text here, including writing the value as "0.87" and changing "carbon" to "C" throughout the manuscript.

11. Table 1: (A) I would change the order of the soil types from "Gelisol – Mollisol Inceptisol" to the order of the assumed soil development, i.e. "Gelisol – Inceptisol – Mollisol" (B) I had to search for the meaning of each parameter in the Material and Methods section and this is cumbersome. Please add a short explanation of all parameters listed in Table 1 either in the header or as a footnote (e.g. Cs – surface C, Cmin – minimum C etc.) (C) Additionally, I did not even find an explanation for all of the parameters: What are the adjusted parameters (Zadj, Z*adj, Zmin_adj?) – what did you do to adjust them? This information needs to be added in the Materials and Methods section. (D) Please add units for the parameters given

We changed the order of the profile types in Table 1. We also added a footnote with definitions for all parameters and units for each parameter. We specifically not that the "adjusted" parameters are adjusted for organic horizon thickness.

12. Table 2: Please add a column with the soil type here as you did in table 1

We revised Table 2 to include a "Soil type" column.

13. Table S1: Profile HCCN2/3 starts at 24 cm soil depth? What happened to the upper 23 cm?

The upper 23 cm is the O horizon (following O'Donnell et al. 2011).